# Comparison of alignment and spondylolysis fracture angle in bilateral and unilateral spondylolysis

Kanta Matsuzawa[1,2]☯*, Tomoyuki Matsui[1], Yoshikazu Azuma[1], Tetsuya Miyazaki[1,2], Machiko Hiramoto[1], Ruo Hashimoto[1], Noriyuki Kida[2], Toru Morihara[1]☯

1 Marutamachi Rehabilitation Clinic, Kyoto, Japan, 2 Department of Biotechnology, Graduate School of Science and Technology, Kyoto Institute of Technology, Kyoto, Japan

☯ These authors contributed equally to this work.

* k.matsuzawa.pt@gmail.com

## Abstract

### Objectives

Spondylolysis occurs bilaterally or unilaterally and bilateral spondylolysis increases the risk of developing isthmic spondylolisthesis. The characteristics of the lumbar lordosis angle (LLA), sacral slope angle (SSA), and spondylolysis fracture angle (SFA) in bilateral spondylolysis compared with those in unilateral spondylolysis have not been clarified. The purpose of this study was to compare the LLA, SSA, and SFA of bilateral and unilateral spondylolysis.

### Materials and methods

Thirty-eight patients with lumbar spondylolysis who visited our clinic for an initial visit and 15 age-matched patients with a chief complaint of low back pain were included as controls. Computed tomography films were used to classify all spondylolysis patients into two groups: those with bilateral fractures (bilateral) and those with unilateral fractures (unilateral). The LLA and SSA were measured using lateral X-ray films and the SFA was measured using computed tomography films.

### Results

The LLA was significantly higher in all spondylolysis patients than in the control group (p = .026). There was no significant difference in SSA between the spondylolysis and control groups (p = .28). The LLA was significantly higher in the bilateral group than in the unilateral group (p = .018). There was no significant difference in SSA between the bilateral and unilateral groups (p = .15). The SFA was significantly lower in the bilateral group than in the unilateral group (p = .024).

### Conclusions

This study suggests that physical therapy for spondylolysis may be considered bilaterally and unilaterally.

**Data Availability Statement:** All relevant data are within the paper and its Supporting Information files.

**Funding:** The authors received no specific funding for this work.

**Competing interests:** The authors declare that no competing interests exist.

## Introduction

Lumbar spondylolysis frequently occurs in adolescent athletes [1], and conservative treatment is commonly performed [2, 3]. Spondylolysis occurs bilaterally or unilaterally and bilateral spondylolysis increases the risk of developing isthmic spondylolisthesis [4]. Using clinical and biomechanical studies, Sairyo et al. [5] reported that unilateral spondylolysis could lead to bilateral spondylolysis. Among spondylolysis who had bony healing with orthotic conservative treatment, maximum patient satisfaction was achieved in 54% of patients [6]. Physical therapy may be important to improve clinical outcome and prevent recurrence [6]. Therefore, we considered that clarifying the characteristics of bilateral spondylolysis compared to those of unilateral spondylolysis may assist in physical therapy to prevent transition to bilateral spondylolysis or isthmic spondylolisthesis and a recurrence.

Abnormal alignment is a physical feature of spondylolysis. A previous study reported that lumbar lordosis angle (LLA) was increased in spondylolysis patients compared to that in healthy controls [7]. Increase in LLA may lead to an increase in the compressive and shearing forces on the pars interarticularis with lumbar extension, which may increase the risk of developing spondylolysis [8]. Sacral slope angle (SSA) is an indicator of the risk of progression from spondylolysis to spondylolisthesis [9, 10].

Spondylolysis fracture angle (SFA) is one of the imaging findings of spondylolysis [11, 12]. Previous studies have reported that SFA was significantly higher in rotation-related sports than in non- rotation-related sports [11, 12]. Sairyo et al. [11] reported that the stresses applied to the pars interarticularis during movement were higher during extension and rotation, and that the stress direction was more coronal during extension than during other movements using the finite element model. Therefore, SFA may infer the mechanical stress that the pars interarticularis is being applied [11, 12]. To the best of our knowledge, the characteristics of LLA, SSA, and SFA in bilateral spondylolysis compared with unilateral spondylolysis have not been clarified. If this can be clarified, we believe that it will provide useful information when considering the contents of physical therapy.

The purpose of this study was to compare the LLA, SSA, and SFA in bilateral and unilateral spondylolysis. We hypothesized that LLA and SSA would be higher and SFA would be lower in bilateral spondylolysis than in unilateral spondylolysis.

## Materials and methods

### Participants

This was a retrospective study. The participants were 38 patients with lumbar spondylolysis who visited our clinic for an initial visit between January 2020 and December 2021 (mean age, 15.32 ± 1.95 years; 32 boys, 6 girls, 27 at L5, 9 at L4, and 3 at L3). Inclusion criteria were as follows: patients who had computed tomography (CT) and X-ray films in the side-lying position for the initial visit, and patients with progressive or terminal stage [12]. Exclusion criteria were as follows: patients with a history of lower back pain at other hospitals, those with recurrent spondylolysis, and those with a history of orthopedic surgery. Fifteen males, age-matched and presenting with lumbar discopathy, excluding those with spondylolysis, were included as controls. Parents gave written informed consent and children gave their assent to participate in the study. This study was performed in accordance with the declaration of Helsinki after obtaining approval from the ethics committee at our institution (Raku-gaku-Rin-01-000100).

CT films were used to classify all spondylolysis cases into two groups: those with bilateral fractures (bilateral) and those with unilateral fractures (unilateral).

## Measurements

The lumbar lordosis angle (LLA) was measured with reference to previous study [13] using lateral X-ray films. First, lines were drawn tangential to the superior endplate of L1 and inferior endplate of L5. Second, perpendicular lines were drawn to each tangent. The LLA was defined as the acute angle formed by the intersection of two perpendicular lines (Fig 1A).

The sacral slope angle (SSA) was measured with reference to previous study [9] using lateral X-ray films. Lines were drawn tangential to the superior endplate of S1 and horizontal. The SSA was defined as the acute angle formed by the two lines (Fig 1B).

The spondylolysis fracture angle (SFA) was measured with reference to previous studies [11, 12] using axial view of the CT films. A previous study [14] reported that spondylolysis originates from the caudal-ventral aspect of the pars interarticularis. Therefore, the SFA was measured using the horizontal section of the most caudal part of the fracture. The SFA was defined as the angle between the line parallel to the posterior cortex of the vertebral body and fracture line, with the direction of the spinous process being positive (Fig 2).

## Statical analysis

Student's t-test was used to compare the LLA and SSA between all spondylolysis and control groups. The mean values of both sides were used for SFA of the bilateral group. The Student's t-test was used to compare the LLA, SSA, and SFA between the bilateral and unilateral groups.

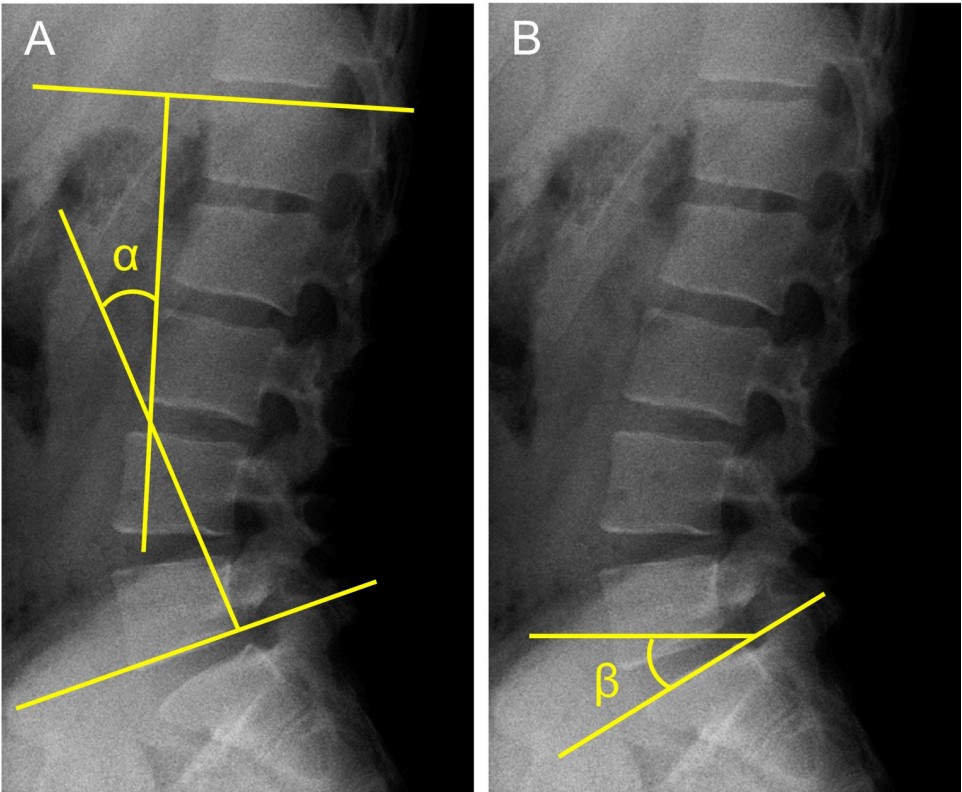

**Fig 1. Measurement of lumbar lordosis angle (LLA) and sacral slope angle (SSA).** (A) First, lines were drawn tangential to the superior endplate of L1 and the inferior endplate of L5 on X-ray films. Second, perpendicular lines were drawn to each tangent. The LLA (α) was defined as the acute angle formed by the intersection of two perpendicular lines. (B) Lines were drawn tangential to the superior endplate of S1 and horizontal on the X-ray films. The SSA (β) was defined as the acute angle formed by the two lines.

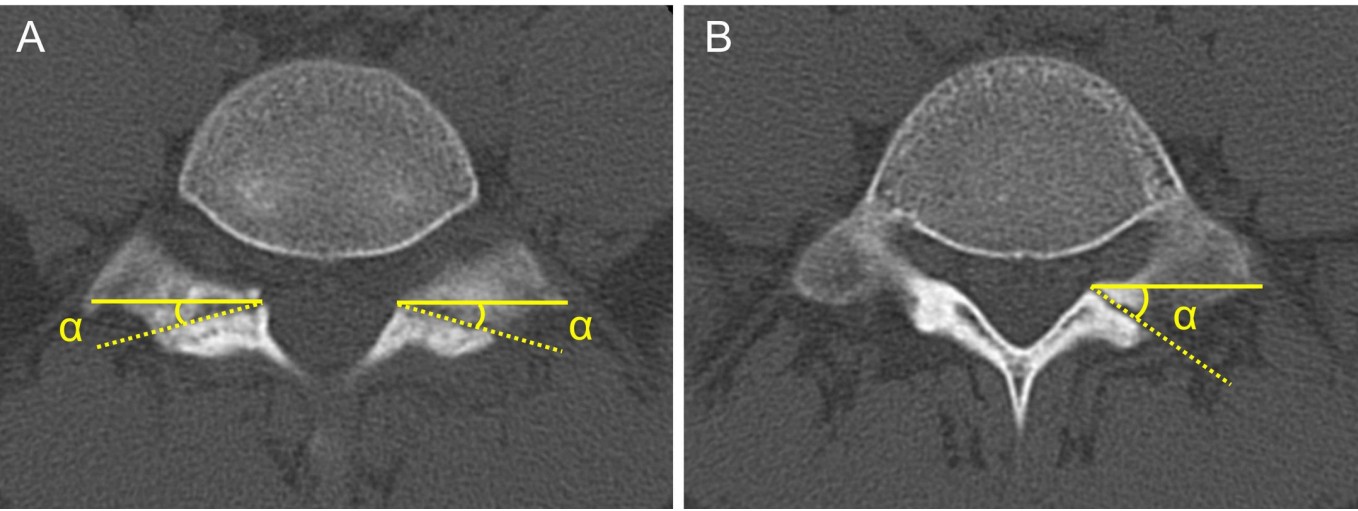

**Fig 2. Measurement of spondylolysis fracture angle (SFA).** The SFA ($\alpha$) was defined as the angle between the line parallel to the posterior cortex of the vertebral body (solid line) and fracture line (dotted line), with the direction of the spinous process being positive. (A) Bilateral spondylolysis. (B) Unilateral spondylolysis.

Pearson's correlation coefficient was used to determine the correlation between SFA and LLA, bilaterally and unilaterally. For all the analyses, level of significance was set at $p < .05$.

## Results

Among the patients with spondylolysis, 23 (61%) had bilateral spondylolysis and 15 (39%) had unilateral spondylolysis. The fracture level of the patients in the bilateral group was the same on both sides. The bilateral group comprised 13 baseball players, four soccer players, three volleyball players, one track and field player, one karate player, and one badminton player. The unilateral group comprised 13 baseball players, one volleyball player, and one track and field player.

The LLA was significantly larger in all the spondylolysis patients than in the control group ($p = .026$). There was no significant difference in the SSA between the spondylolysis and control groups ($p = .28$) (Table 1). The LLA was significantly larger in the bilateral group than in the unilateral group ($p = .018$). There was no significant difference in the SSA between the bilateral and unilateral groups ($p = .15$). The SFA was significantly smaller in the bilateral group than in the unilateral group ($p = .024$) (Table 2). There was no significant correlation between SFA and LLA [bilateral ($p = .072$), unilateral ($p = .76$)] (Fig 3).

**Table 1. Measured data for all spondylolysis and control group.**

|  | All spondylolysis | Control | P value[*] |
|---|---|---|---|
| LLA (°) | 21.07 ± 8.71 | 15.39 ± 6.17 | 0.026 |
| SSA (°) | 29.31 ± 7.22 | 27.05 ± 5.47 | 0.28 |
| SFA (°) | 14.28 ± 11.18 | – | – |

[a]Values are means ± SD

[b]LLA, lumbar lordosis angle; SSA, sacral slope angle; SFA, spondylolysis fracture angle;

[*] Student's t-test.

**Table 2. Measured data for all patient in each group.**

|  | Bilateral | Unilateral | P value[*] |
|---|---|---|---|
| LLA (˚) | 23.73 ± 9.20 | 16.99 ± 6.16 | 0.018 |
| SSA (˚) | 30.66 ± 8.14 | 27.23 ± 5.08 | 0.15 |
| SFA (˚) | 15.35 ± 8.82 | 22.44 ± 9.47 | 0.024 |

[a]Values are means ± SD

[b]LLA, lumbar lordosis angle; SSA, sacral slope angle; SFA, spondylolysis fracture angle;

[*] Student's t-test.

## Discussion

This study compared alignment using X-ray films and SFA using CT films in bilateral and unilateral spondylolysis. We measured LLA and SSA as alignment; LLA was significantly larger in all the spondylolysis patients than in the control group, and the LLA was significantly larger in bilateral than unilateral spondylolysis. Increase in LLA may lead to an increase in compressive and shearing forces on the pars interarticularis with lumbar extension [8]. Therefore, this study suggested that the bilateral group may have experienced increased stress during extension compared to unilateral group. There were no significant differences in the SSA between the spondylolysis and control groups, and between bilateral and unilateral groups. Lumbar lordosis is related to the alignment of the upper spine and pelvis. Therefore, LLA in the bilateral group may be related to the upper spine.

In this study, the SFA was significantly smaller in the bilateral group than in the unilateral group. A previous study reported that the stress direction in the horizontal section was more

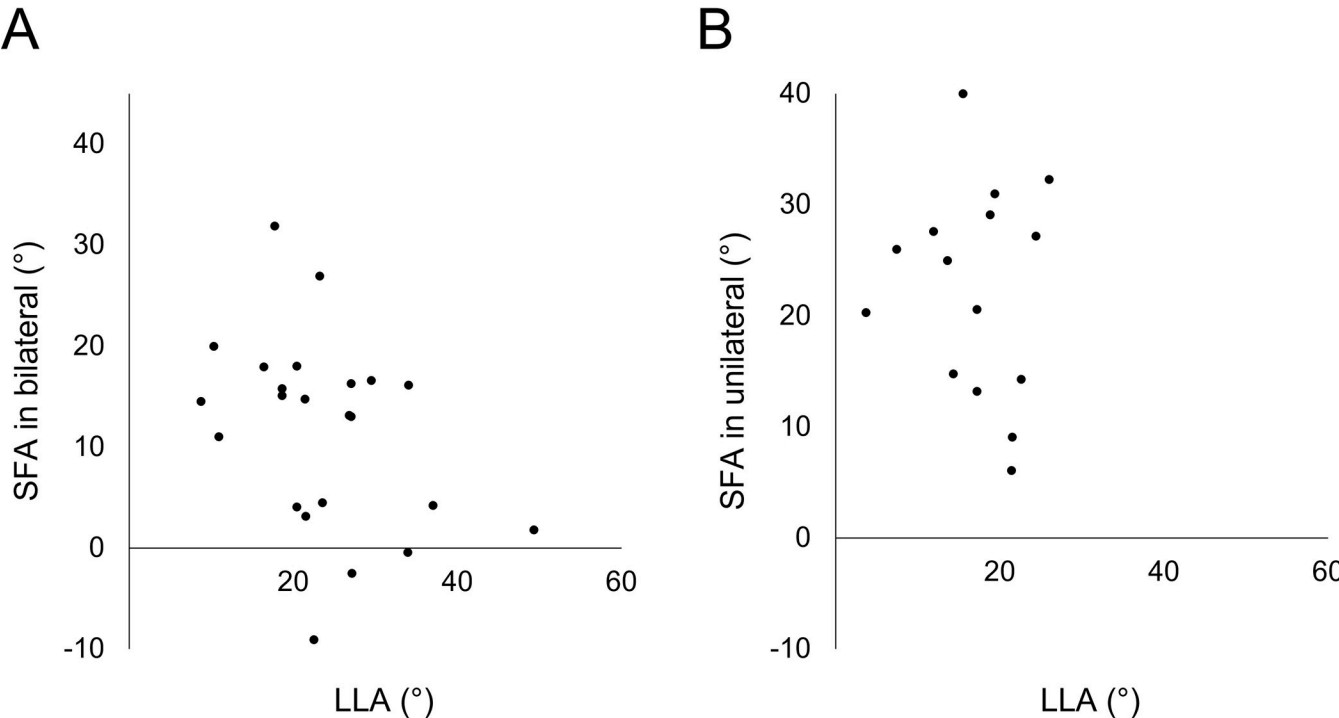

**Fig 3.** Scatter plot of spondylolysis fracture angle (SFA) and lumbar lordosis angle (LLA) in bilateral spondylolysis (A) and unilateral spondylolysis (B). There was no significant correlation between SFA and LLA [bilateral (p = .072), unilateral (p = .76)].

coronal during extension than during other movements [11]. Therefore, this study suggests that excessive extension stress may have been applied in the bilateral group compared to in the unilateral group. There was no significant correlation between SFA and LLA in this study. Therefore, further investigation is necessary because factors other than LLA may be involved in SFAs.

Taking the results of this study into consideration, we believe that bilateral spondylolysis needs more improvement in lumbar lordosis than unilateral spondylolysis. In spondylolysis, it is necessary to acquire physical functions and movements to reduce extension and rotation stress, and we believe that it is necessary to focus more on extension stress in bilateral spondylolysis and on rotation stress in unilateral spondylolysis. In the future, standing upright posture alignment, physical functions, and movements should be clarified.

This study has two limitations. First, there was a bias toward sports. However, bilateral and unilateral spondylolysis were observed in the same sports. Second, it was not known whether the bilateral patients had migrated unilaterally. Therefore, this study excluded this factor as much as possible by including a criteria that had never been to another hospital for low back pain.

## Conclusion

In spondylolysis, bilateral spondylolysis showed a larger LLA and smaller SFA than unilateral spondylolysis. This study suggests that physical therapy for spondylolysis may be considered bilaterally or unilaterally.

## Supporting information

**S1 File. Data set in all participants.**
(XLSX)

## Acknowledgments

We would like to thank Editage (www.editage.com) for English language editing.

## Author Contributions

**Data curation:** Kanta Matsuzawa, Machiko Hiramoto, Ruo Hashimoto.

**Formal analysis:** Kanta Matsuzawa, Machiko Hiramoto, Ruo Hashimoto, Noriyuki Kida.

**Investigation:** Yoshikazu Azuma.

**Methodology:** Kanta Matsuzawa, Tomoyuki Matsui, Tetsuya Miyazaki, Noriyuki Kida.

**Project administration:** Toru Morihara.

**Validation:** Yoshikazu Azuma, Tetsuya Miyazaki.

**Writing – original draft:** Kanta Matsuzawa.

**Writing – review & editing:** Tomoyuki Matsui, Toru Morihara.

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
