## [Decision Letter · Decision Letter 0]

25 Aug 2022

PONE-D-22-11206Comparison of alignment and spondylolysis fracture angle in bilateral and unilateral spondylolysisPLOS ONE

Dear Dr. Matsuzawa,

Thank you for submitting your manuscript to PLOS ONE. After careful consideration, we feel that it has merit but does not fully meet PLOS ONE’s publication criteria as it currently stands. Therefore, we invite you to submit a revised version of the manuscript that addresses the points raised during the review process.

I would like to sincerely apologise for the delay you have incurred with your submission. It has been exceptionally difficult to secure reviewers to evaluate your study. We have now received two completed reviews; the comments are available below. The reviewers have raised significant scientific concerns about the study that need to be addressed in a revision. Please revise the manuscript to address all the reviewer's comments in a point-by-point response in order to ensure it is meeting the journal's publication criteria. Please note that the revised manuscript will need to undergo further review, we thus cannot at this point anticipate the outcome of the evaluation process.

We look forward to receiving your revised manuscript.

Kind regards,

Miquel Vall-llosera Camps

Senior Editor

PLOS ONE

Journal Requirements:

“NO”

“NO authors have competing interests”

Reviewers' comments:

Reviewer's Responses to Questions

**Comments to the Author**

1. Is the manuscript technically sound, and do the data support the conclusions?

Reviewer #1: Partly

Reviewer #2: Yes

2. Has the statistical analysis been performed appropriately and rigorously? 

Reviewer #1: Yes

Reviewer #2: Yes

3. Have the authors made all data underlying the findings in their manuscript fully available?

Reviewer #1: Yes

Reviewer #2: Yes

4. Is the manuscript presented in an intelligible fashion and written in standard English?

Reviewer #1: No

Reviewer #2: Yes

5. Review Comments to the Author

Reviewer #1: General comments

The purpose of this study was to report the characteristics of X-ray measurement between the unilateral and bilateral spondylolysis patients along with comparison between the controls.

It is truly important to reveal the characteristics of the pathology of the spondylolysis, as it may affect the methodology of physical treatment, depending on the physical feature. I understand the results that the spondylolysis cases had larger LLA compared to the controls and bilateral cases showed smaller SFA compared to the unilateral cases. However, there seems to be many flaws that should be revised and explained more carefully.

I have also noticed many grammatical errors and expressions which needs to be revised. I suggest you to have the native speaker involved or have the manuscript edited by a language service again. Therefore, I will leave it to the editor-in-chief about the final decision, however, there seems a lot of points that needs to be revised.

Specific comments

Title

OK.

Abstract

Line 20

Why is “Spondylolysis fracture angle” capitalized? Not a proper noun and does not have to be capitalized.

Line 28

“The” should not be capitalized if it is not mentioning about the proper noun or beginning the sentence. See many of these errors regarding the “capitalization”. Please check the manuscript thoroughly.

Line 31

If they are comparing only the SSA, this sentence is stating about a singular issue. Therefore, the sentence should be revised as a singular manner. “There was no significant difference in…”

Introduction

Lines 40-41

conservative treatment is commonly performed

Line 44

Isn’t it evident that physical therapy is important to improve clinical outcomes and to prevent recurrence? You should emphasize the importance of physical therapy, if you are to conclude the possibility of difference in approach between the unilateral and bilateral patients.

Lines 46-47

Need to revise the expression of parallel structure.

Line 48

Just the alignment? Not “abnormal” alignment? Alignment can be assessed in many pathologies, not just in spondylolysis. Needs to revise the expression for readability and delve in deeper to tell the readers why the alignment is an important physical feature among the patients with spondylolysis.

Line 50

“Increasing the LLA” sounds odd. No one increases the LLA. LLA is increased as a result of mechanical stress and pathology of spondylolysis. “Increase in the LLA” may be proper. If the expression will be repeating the use of word “increase”, then the whole sentence should be revised for readability.

Line 55

Is “twist sports” appropriate phrase? I suppose the previous study have presented as “rotation-related sports”.

Line 57

Direction cannot be expressed as “section”. “stress distributed to the horizontal direction” or any kind of expression resembling this might be appropriate.

This can also be said in the Discussion (eg. Line 149).

Materials and Methods

Line 68

I suppose there are more better ways to express than to start a sentence with “First”. A bit annoying. Just by saying “This study was conducted as a retrospective study” sound enough.

Line 70

mean

It is not a beginning of a sentence.

Line 74

What kind of orthopedic surgery? Any type and location of the orthopedic surgery?

Was only age-matched? How about sex?

Line 75

If possible, can you present the specific diagnosis of the patients who were evaluated as controls? “Low back pain” is just one of symptoms and includes many pathologies. I suppose to reveal the diagnosis is very helpful to the readers.

Lines 78-79

Method of obtaining the CT-scan images should be more specific. As the previous report from Nagamoto et al. have mentioned, there are two ways to measure SFA. Measuring the SFA by the images from axial view and plane parallel to the pars may affect the results of the measured angle.

Lines 89

perpendicular

Not a beginning of a sentence or a proper noun.

Lines 100-101

“angle between the horizontal and fracture lines” is not defining the measurement method clearly. What is the “horizontal line”? If it was a line parallel to the posterior cortex of the vertebral body, then it must be stated specifically to let the readers understand the methods more clear and simple.

Lines 105-106

I suppose you have given this figure to let the readers understand the bilateral or unilateral spondylolysis is, regardless of participating sports. If so, “baseball” player or not is unnecessary information.

Results

The expression of measured angle values is not appropriate. The angles should not be expressed as “high” or “low”. Maybe “large” or “small” is more suitable. This could also be said to the expression in the Discussion.

Lines 116 & 119

“bilateral sports” and “unilateral sports” sounds odd. This may be able to interpret as a sport which you use your hand or foot unilaterally or bilaterally. Meant to express those with “bilateral spondylolysis” and “unilateral spondylolysis”? If so, needs to revise the expression

In addition, among all the bilaterally fractured players, were all the fractures occurred at the same level of vertebra? Or were fractures appeared in different level of vertebrae? I suggest to define it more clearly. If those with bilaterally fractured but happened to have them in different level of vertebrae, SFA may differ according to how the mechanical stress concentrated to the pars, including what kind of sports they participated, which may affect the results of the angle.

Lines 124-125

How was the average angle of SFA measured in bilateral groups? Were they calculated as an average between both sides? If so, I suppose there were some cases that showed horizontal-type on one side and rotation-type fracture on the other side. This difference may affect the results if you are to discuss this matter as an “average”. This could relate to the issue I have mentioned above. This must be clarified and stated.

Line 138

lumbar lordosis angle

Discussion

Lines 141-147

I understood that the LLA was larger in all spondylolysis group compared to the control and larger in bilateral compared to unilateral. However, how would you consider about the results of SSA, which did not differ between any of the comparison? If you are to conclude that bilateral group may have more of an extension stress than unilateral group, then SSA could have been larger in bilateral group. Can you discuss more specifically according to your results?

Line 148

Repeating the same expression consecutively as a beginning of the paragraphs is annoying. Needs to revise the style of expression.

Line 151

The word “attack” sounds improper here. Someone or something will not “attack” pars. Mechanical stress occurs from his/her own exercise.

Line 158

alignment in standing upright posture

Conclusions

Can you conclude that physical therapy should be considered between bilateral and unilateral cases, just by the results of this study? Same can be said to the abstract. This study just revealed the difference of X-ray and CT-scan measurement, not anything related to the mechanism of development or physical condition. The measurement which the authors have revealed are only the “results” of the spondylolysis. If the authors would like to correlate the results of this study to physical therapy, the authors need to delve in deeper in the discussion.

Tables and Figure Legends

Table 1

All spondylolysis

“spondylolysis” is not a proper noun.

References

Are all the references written in the format according to the instruction by the journal?

Name of the journal is a proper noun. Every word has to be capitalized except for the conjunction. Journal No.1 must be “Journal of Orthopaedic Science”, journal No.2 must be “Sports Medicine and Arthroscopy Review”, journal No.3 must be “Physical Therapy in Sport”, journal No.4 must be “American Journal of Sports Medicine”.

Also, above mentioned names of the journals are expressed in full name, even though the journal No 6 is abbreviated.

The authors must inspect the manuscript thoroughly and check the grammatical errors.

Reviewer #2: Very interesting and poorly known subject.

I have nevertheless certain requirements:

The authors should more analyze the absence of difference regarding SSA between the groups, especially between the unilateral fracture group and the bilateral fracture group.

Did the measurements of the LLA and SSA have been carried out on standing position? the authors should precise it.

Figure 3 should be more explained because the graphic remains unclear.

Introduction part, methods part and results part are consistent without needing modification.

In the discussion part, the authors should more argue their hypothesis as concern the occurrence of isthmic unilateral fracture rather than bilateral isthmic fracture. It would be possibly relevant to add a schematic figure explaining the relationship between SFA, uni or bilateral isthmic fracture and type of sport movement leading to isthmic stress fracture.

6. PLOS authors have the option to publish the peer review history of their article (what does this mean?). If published, this will include your full peer review and any attached files.

Reviewer #1: No

Reviewer #2: No

---

## [Author Response · Author response to Decision Letter 0]

5 Sep 2022

September 1, 2022

Editorial Board

PLOS ONE

Manuscript Number: PONE-D-22-11206

“Comparison of alignment and spondylolysis fracture angle in bilateral and unilateral spondylolysis” by Kanta Matsuzawa

Dear Editor:

Thank you for your letter regarding our submission. We are grateful for the detailed feedback provided by the reviewers, which we feel has helped us significantly improve the paper. Attached are our point-by-point responses to the reviewers’ comments and our revised manuscript, which we hope will now meet your approval. We believe that our revisions have addressed the issues raised by the reviewers and trust that the manuscript is now suitable for publication in PLOS ONE.

Journal Requirements:

→We have ensured the needful.

2. We note that you have indicated that data from this study are available upon request.

→We have now uploaded the data as the Supporting Information files.

Data Availability Statement: All relevant data are included in the manuscript and the Supporting Information files.

3. Thank you for stating the following financial disclosure:“NO”

→Thank you for pointing this out. This has been revised appropriately.

Funding: The authors received no specific funding for this work.

4. Thank you for stating the following in your Competing Interests section: “NO authors have competing interests”

→Thank you for pointing this out. This has been appropriately revised.

Competing interests: The authors declare that no competing interests exist.

Thank you again for your thoughtful comments, and we look forward to hearing from you soon.

Sincerely,

Kanta Matsuzawa 

Reviewer #1:

General comments

The purpose of this study was to report the characteristics of X-ray measurement between the unilateral and bilateral spondylolysis patients along with comparison between the controls.

It is truly important to reveal the characteristics of the pathology of the spondylolysis, as it may affect the methodology of physical treatment, depending on the physical feature. I understand the results that the spondylolysis cases had larger LLA compared to the controls and bilateral cases showed smaller SFA compared to the unilateral cases. However, there seems to be many flaws that should be revised and explained more carefully.

I have also noticed many grammatical errors and expressions which needs to be revised. I suggest you to have the native speaker involved or have the manuscript edited by a language service again. Therefore, I will leave it to the editor-in-chief about the final decision, however, there seems a lot of points that needs to be revised.

→Thank you for the suggestion. The manuscript has been revised by a professional academic editor whose first language is English and who is associated with Editage.

Specific comments

Abstract

1　Line 20-Why is “Spondylolysis fracture angle” capitalized? Not a proper noun and does not have to be capitalized.

→Thank you for the suggestion. We have revised this appropriately (Lines 19- ).

The characteristics of the lumbar lordosis angle (LLA), sacral slope angle (SSA), and spondylolysis fracture angle (SFA) in bilateral spondylolysis compared with those in unilateral spondylolysis have not been clarified.

2　Line 28-“The” should not be capitalized if it is not mentioning about the proper noun or beginning the sentence. See many of these errors regarding the “capitalization”. Please check the manuscript thoroughly.

→Apologies for the errors. These have been revised appropriately at all applicable places. (Lines 27- ).

The LLA and SSA were measured using lateral X-ray films and the SFA was measured using computed tomography films.

3　Line 31-If they are comparing only the SSA, this sentence is stating about a singular issue. Therefore, the sentence should be revised as a singular manner. “There was no significant difference in…”

→We have revised this as suggested (Lines 31-, 124-, 125- ).

・There was no significant difference in SSA between the spondylolysis and control groups (p = .28).

・There was no significant difference in the SSA between the spondylolysis and control groups (p = .28) (Table 1).

・There was no significant difference in the SSA between the bilateral and unilateral groups (p = .15).

Introduction

4　Lines 40-41-conservative treatment is commonly performed

→Thank you for the suggestion. We have revised as suggested (Lines 40- ).

Lumbar spondylolysis frequently occurs in adolescent athletes [1], and conservative treatment is commonly performed [2, 3].

5　Line 44-Isn’t it evident that physical therapy is important to improve clinical outcomes and to prevent recurrence? You should emphasize the importance of physical therapy, if you are to conclude the possibility of difference in approach between the unilateral and bilateral patients.

→We have added the following text (Line 44- ): Among spondylolysis who had bony healing with orthotic conservative treatment, maximum patient satisfaction was achieved in 54% of patients [6].

Lumbar spondylolysis frequently occurs in adolescent athletes [1], and conservative treatment is commonly performed [2, 3]. Spondylolysis occurs bilaterally or unilaterally and bilateral spondylolysis increases the risk of developing isthmic spondylolisthesis [4]. Using clinical and biomechanical studies, Sairyo et al. [5] reported that unilateral spondylolysis could lead to bilateral spondylolysis. Among spondylolysis who had bony healing with orthotic conservative treatment, maximum patient satisfaction was achieved in 54% of patients [6]. Physical therapy may be important to improve clinical outcome and prevent recurrence [6]. Therefore, we considered that clarifying the characteristics of bilateral spondylolysis compared to those of unilateral spondylolysis may assist in physical therapy to prevent transition to bilateral spondylolysis or isthmic spondylolisthesis and a recurrence.

6. El Rassi G, Takemitsu M, Glutting J, Shah SA. Effect of sports modification on clinical outcome in children and adolescent athletes with symptomatic lumbar spondylolysis. American Journal of Physical Medicine and Rehabilitation. 2013;92(12):1070-4. Epub 2013/10/22. doi: 10.1097/PHM.0b013e318296da7e. PubMed PMID: 24141103.

6　Lines 46-47-Need to revise the expression of parallel structure.

→We have revised this sentence (Line 50- ).

Therefore, we considered that clarifying the characteristics of bilateral spondylolysis compared to those of unilateral spondylolysis may assist in physical therapy to prevent transition to bilateral spondylolysis or isthmic spondylolisthesis and a recurrence.

7　Line 48-Just the alignment? Not “abnormal” alignment? Alignment can be assessed in many pathologies, not just in spondylolysis. Needs to revise the expression for readability and delve in deeper to tell the readers why the alignment is an important physical feature among the patients with spondylolysis.

→We have revised as suggested (Line 54- ).

Abnormal alignment is a physical feature of spondylolysis.

8　Line 50-“Increasing the LLA” sounds odd. No one increases the LLA. LLA is increased as a result of mechanical stress and pathology of spondylolysis. “Increase in the LLA” may be proper. If the expression will be repeating the use of word “increase”, then the whole sentence should be revised for readability.

→We have revised this sentence (Lines 52- ).

Increase in LLA may lead to an increase in the compressive and shearing forces on the pars interarticularis with lumbar extension, which may increase the risk of developing spondylolysis [8].

9　Line 55-Is “twist sports” appropriate phrase? I suppose the previous study have presented as “rotation-related sports”.

→We have revised as suggested (Lines 57- ).

Previous studies have reported that SFA was significantly higher in rotation-related sports than in non- rotation-related sports [11, 12].

11. Sairyo K. Spondylolysis Fracture Angle in Children and Adolescents on CT Indicates the Facture Producing Force Vector : A Biomechanical Rationale. The Internet Journal of Spine Surgery. 2005;1(2):2.

12. Nagamoto H, Abe M, Konashi Y, Kimura R, Takahashi M, Oizumi A. Rotation-related sports players demonstrate rotation-type lumbar spondylolysis fracture angle and decreased hip internal rotation range of motion. Journal of Orthopaedics. 2021;28:101-6. Epub 2021/12/14. doi: 10.1016/j.jor.2021.11.010. PubMed PMID: 34898928; PubMed Central PMCID: PMCPMC8640617.

10　Line 57-Direction cannot be expressed as “section”. “stress distributed to the horizontal direction” or any kind of expression resembling this might be appropriate.

This can also be said in the Discussion (eg. Line 149).

→We have revised this sentence and the Discussion (Line 58-, ).

・Sairyo et al. [11] reported that the stresses applied to the pars interarticularis during movement were higher during extension and rotation, and that the stress direction was more coronal during extension than during other movements using the finite element model.

・A previous study reported that the stress direction in the horizontal section was more coronal during extension than during other movements [11].

11. Sairyo K. Spondylolysis Fracture Angle in Children and Adolescents on CT Indicates the Facture Producing Force Vector : A Biomechanical Rationale. The Internet Journal of Spine Surgery. 2005;1(2):2.

Materials and Methods

11　Line 68-I suppose there are more better ways to express than to start a sentence with “First”. A bit annoying. Just by saying “This study was conducted as a retrospective study” sound enough.

→Thank you for the suggestion. We have removed "First" (Line 70- ).

This was a retrospective study.

12　Line 70-mean　It is not a beginning of a sentence.

→We have revised as suggested (Lines 70- ).

The participants were 38 patients with lumbar spondylolysis who visited our clinic for an initial visit between January 2020 and December 2021 (mean age, 15.32 ± 1.95 years; 32 boys, 6 girls, 27 at L5, 9 at L4, and 3 at L3).

13　Line 74-What kind of orthopedic surgery? Any type and location of the orthopedic surgery? Was only age-matched? How about sex?

→Thank you for these questions. Based on these, we have revised the participants in the Materials and methods section (Lines 76- ).

Fifteen males, age-matched and presenting with lumbar discopathy, excluding those with spondylolysis, were included as controls.

14　Line 75-If possible, can you present the specific diagnosis of the patients who were evaluated as controls? “Low back pain” is just one of symptoms and includes many pathologies. I suppose to reveal the diagnosis is very helpful to the readers.

→Thank you for this valuable suggestion. Accordingly, we have revised the participants in the Materials and methods section (Lines 76- ).

Fifteen males, age-matched and presenting with lumbar discopathy, excluding those with spondylolysis, were included as controls.

15　Lines 78-79-Method of obtaining the CT-scan images should be more specific. As the previous report from Nagamoto et al. have mentioned, there are two ways to measure SFA. Measuring the SFA by the images from axial view and plane parallel to the pars may affect the results of the measured angle.

→Thank you for this suggestion. This study used axial view of the CT films. We have included the relevant details (Lines 99- ).

The spondylolysis fracture angle (SFA) was measured with reference to previous studies [11, 12] using axial view of the CT films.

11. Sairyo K. Spondylolysis Fracture Angle in Children and Adolescents on CT Indicates the Facture Producing Force Vector : A Biomechanical Rationale. The Internet Journal of Spine Surgery. 2005;1(2):2.

12. Nagamoto H, Abe M, Konashi Y, Kimura R, Takahashi M, Oizumi A. Rotation-related sports players demonstrate rotation-type lumbar spondylolysis fracture angle and decreased hip internal rotation range of motion. Journal of Orthopaedics. 2021;28:101-6. Epub 2021/12/14. doi: 10.1016/j.jor.2021.11.010. PubMed PMID: 34898928; PubMed Central PMCID: PMCPMC8640617.

16　Lines 89-perpendicular　Not a beginning of a sentence or a proper noun.

→Apologies for this error. We have revised as suggested (Lines 91- ).

Second, perpendicular lines were drawn to each tangent.

17　Lines 100-101-“angle between the horizontal and fracture lines” is not defining the measurement method clearly. What is the “horizontal line”? If it was a line parallel to the posterior cortex of the vertebral body, then it must be stated specifically to let the readers understand the methods more clear and simple.

→Thank you for pointing this out. We have revised the measurement details and legend of Fig 2 (Lines 102- ).

The SFA was defined as the angle between the line parallel to the posterior cortex of the vertebral body and fracture line, with the direction of the spinous process being positive (Fig 2).

Fig 2. Measurement of spondylolysis fracture angle (SFA)

The SFA (α) was defined as the angle between the line parallel to the posterior cortex of the vertebral body (solid line) and fracture line (dotted line), with the direction of the spinous process being positive. (A) Bilateral spondylolysis. (B) Unilateral spondylolysis.

18　Lines 105-106-I suppose you have given this figure to let the readers understand the bilateral or unilateral spondylolysis is, regardless of participating sports. If so, “baseball” player or not is unnecessary information.

→We have revised as suggested (Lines 106- ).

Fig 2. Measurement of spondylolysis fracture angle (SFA)

The SFA (α) was defined as the angle between the line parallel to the posterior cortex of the vertebral body (solid line) and fracture line (dotted line), with the direction of the spinous process being positive. (A) Bilateral spondylolysis. (B) Unilateral spondylolysis.

Results

19　The expression of measured angle values is not appropriate. The angles should not be expressed as “high” or “low”. Maybe “large” or “small” is more suitable. This could also be said to the expression in the Discussion.

→Thank you for pointing this out. We have revised appropriately (Lines 123-, 125-, 127-, 146-, 154-, 171- ).

Results

・The LLA was significantly larger in all the spondylolysis patients than in the control group (p = .026).

・The LLA was significantly larger in the bilateral group than in the unilateral group (p = .018).

・The SFA was significantly smaller in the bilateral group than in the unilateral group (p = .024) (Table 2).

Discussion

・We measured LLA and SSA as alignment; LLA was significantly higher in all the spondylolysis patients than in the control group, and the LLA was significantly larger in bilateral than unilateral spondylolysis.

・In this study, the SFA was significantly smaller in the bilateral group than in the unilateral group. A previous study reported that the stress direction in the horizontal section was more coronal during extension than during other movements [11].

Conclusion

In spondylolysis, bilateral spondylolysis showed a larger LLA and smaller SFA than unilateral spondylolysis.

20　Lines 116 & 119-“bilateral sports” and “unilateral sports” sounds odd. This may be able to interpret as a sport which you use your hand or foot unilaterally or bilaterally. Meant to express those with “bilateral spondylolysis” and “unilateral spondylolysis”? If so, needs to revise the expression. In addition, among all the bilaterally fractured players, were all the fractures occurred at the same level of vertebra? Or were fractures appeared in different level of vertebrae? I suggest to define it more clearly. If those with bilaterally fractured but happened to have them in different level of vertebrae, SFA may differ according to how the mechanical stress concentrated to the pars, including what kind of sports they participated, which may affect the results of the angle.

→We have revised the first paragraph of the Results (Line 110- ).

Among the patients with spondylolysis, 23 (61%) had bilateral spondylolysis and 15 (39%) had unilateral spondylolysis. The fracture level of the patients in the bilateral group was the same on both sides. The bilateral group comprised 13 baseball players, four soccer players, three volleyball players, one track and field player, one karate player, and one badminton player. The unilateral group comprised 13 baseball players, one volleyball player, and one track and field player.

21　Lines 124-125-How was the average angle of SFA measured in bilateral groups? Were they calculated as an average between both sides? If so, I suppose there were some cases that showed horizontal-type on one side and rotation-type fracture on the other side. This difference may affect the results if you are to discuss this matter as an “average”. This could relate to the issue I have mentioned above. This must be clarified and stated.

→The SFA of the bilateral group was the mean of both sides. We have described this in the statistical analysis of the Materials and Methods section of the revised manuscript. (Line 113- ).

The mean values of both sides were used for SFA of the bilateral group.

22　Line 138-lumbar lordosis angle

→We have revised as suggested (Lines 141- ).

Fig 3. Scatter plot of spondylolysis fracture angle (SFA) and lumbar lordosis angle (LLA) in bilateral spondylolysis (A) and unilateral spondylolysis (B).

Discussion

23　Lines 141-147-I understood that the LLA was larger in all spondylolysis group compared to the control and larger in bilateral compared to unilateral. However, how would you consider about the results of SSA, which did not differ between any of the comparison? If you are to conclude that bilateral group may have more of an extension stress than unilateral group, then SSA could have been larger in bilateral group. Can you discuss more specifically according to your results?

→Thank you for your questions. We have added text to the Discussion section to clarify this further. (Line 151- ).

There were no significant differences in the SSA between the spondylolysis and control groups, and between bilateral and unilateral groups. Lumbar lordosis is related to the alignment of the upper spine and pelvis. Therefore, LLA in the bilateral group may be related to the upper spine.

24　Line 148-Repeating the same expression consecutively as a beginning of the paragraphs is annoying. Needs to revise the style of expression.

→We have revised as suggested (Lines 145- ).

This study compared alignment using X-ray films and SFA using CT films in bilateral and unilateral spondylolysis. We measured LLA and SSA as alignment; LLA was significantly higher in all the spondylolysis patients than in the control group, and the LLA was significantly larger in bilateral than unilateral spondylolysis. Increasing the LLA may lead to an increase in compressive and shearing forces on the pars interarticularis with lumbar extension [8]. Therefore, this study suggested that the bilateral group may have experienced increased stress during extension compared to unilateral group. There were no significant differences in the SSA between the spondylolysis and control groups, and between bilateral and unilateral groups. Lumbar lordosis is related to the alignment of the upper spine and pelvis. Therefore, LLA in the bilateral group may be related to the upper spine.

In this study, the SFA was significantly smaller in the bilateral group than in the unilateral group. A previous study reported that the stress direction in the horizontal section was more coronal during extension than during other movements [11]. Therefore, this study suggests that excessive extension stress may have been applied in the bilateral group compared to in the unilateral group. There was no significant correlation between SFA and LLA in this study. Therefore, further investigation is necessary because factors other than LLA may be involved in SFAs.

25　Line 151-The word “attack” sounds improper here. Someone or something will not “attack” pars. Mechanical stress occurs from his/her own exercise.

→Thank you for pointing this out. We have revised this appropriately (Lines 155- ).

Therefore, this study suggests that excessive extension stress may have been applied in the bilateral group compared to in the unilateral group.

26　Line 158-alignment in standing upright posture

→We have revised as suggested (Lines 164- ).

In the future, standing upright posture alignment, physical functions, and movements should be clarified.

Conclusions

27　Can you conclude that physical therapy should be considered between bilateral and unilateral cases, just by the results of this study? Same can be said to the abstract. This study just revealed the difference of X-ray and CT-scan measurement, not anything related to the mechanism of development or physical condition. The measurement which the authors have revealed are only the “results” of the spondylolysis. If the authors would like to correlate the results of this study to physical therapy, the authors need to delve in deeper in the discussion.

→We have modified the expression appropriately (Line 172- ).

In spondylolysis, bilateral spondylolysis showed a larger LLA and smaller SFA than unilateral spondylolysis. This study suggests that physical therapy for spondylolysis may be considered bilaterally or unilaterally.

Tables and Figure Legends

28　Table 1　All spondylolysis　“spondylolysis” is not a proper noun.

→We have revised as suggested (Lines 131- ).

Table 1. Measured data for all spondylolysis and control group

 All spondylolysis Control P value＊

LLA (°) 21.07 ± 8.71 15.39 ± 6.17 0.026

SSA (°) 29.31 ± 7.22 27.05 ± 5.47 0.28

SFA (°) 14.28 ± 11.18 － －

aValues are means ± SD

bLLA, lumbar lordosis angle; SSA, sacral slope angle; SFA, spondylolysis fracture angle; * Student’s t-test.

References

29　Are all the references written in the format according to the instruction by the journal?

Name of the journal is a proper noun. Every word has to be capitalized except for the conjunction. Journal No.1 must be “Journal of Orthopaedic Science”, journal No.2 must be “Sports Medicine and Arthroscopy Review”, journal No.3 must be “Physical Therapy in Sport”, journal No.4 must be “American Journal of Sports Medicine”.

Also, above mentioned names of the journals are expressed in full name, even though the journal No 6 is abbreviated.

The authors must inspect the manuscript thoroughly and check the grammatical errors.

→Thank you for pointing this out. We have formatted the reference list (Lines - ). The text has been corrected by a professional medical editor whose first language is English and who is associated with Editage.

1. Sakai T, Sairyo K, Suzue N, Kosaka H, Yasui N. Incidence and etiology of lumbar spondylolysis: review of the literature. Journal of Orthopaedic Science : official journal of the Japanese Orthopaedic Association. 2010;15(3):281-8. Epub 2010/06/19. doi: 10.1007/s00776-010-1454-4. PubMed PMID: 20559793.

2. Randall RM, Silverstein M, Goodwin R. Review of Pediatric Spondylolysis and Spondylolisthesis. Sports Medicine and Arthroscopy Review. 2016;24(4):184-7. Epub 2016/11/05. doi: 10.1097/JSA.0000000000000127. PubMed PMID: 27811518.

3. Grazina R, Andrade R, Santos FL, Marinhas J, Pereira R, Bastos R, et al. Return to play after conservative and surgical treatment in athletes with spondylolysis: A systematic review. Physical Therapy in Sport : official journal of the Association of Chartered Physiotherapists in Sports Medicine. 2019;37:34-43. Epub 2019/03/04. doi: 10.1016/j.ptsp.2019.02.005. PubMed PMID: 30826586.

4. Cooper G. Non-Operative Treatment of the Lumbar Spine. 2015.

5. Sairyo K, Katoh S, Sasa T, Yasui N, Goel VK, Vadapalli S, et al. Athletes with unilateral spondylolysis are at risk of stress fracture at the contralateral pedicle and pars interarticularis: a clinical and biomechanical study. The American Journal of Sports Medicine. 2005;33(4):583-90. Epub 2005/02/22. doi: 10.1177/0363546504269035. PubMed PMID: 15722292.

6. El Rassi G, Takemitsu M, Glutting J, Shah SA. Effect of sports modification on clinical outcome in children and adolescent athletes with symptomatic lumbar spondylolysis. American Journal of Physical Medicine and Rehabilitation. 2013;92(12):1070-4. Epub 2013/10/22. doi: 10.1097/PHM.0b013e318296da7e. PubMed PMID: 24141103.

7. Roussouly P, Gollogly S, Berthonnaud E, Labelle H, Weidenbaum M. Sagittal alignment of the spine and pelvis in the presence of L5-s1 isthmic lysis and low-grade spondylolisthesis. Spine (Phila Pa 1976). 2006;31(21):2484-90. Epub 2006/10/07. doi: 10.1097/01.brs.0000239155.37261.69. PubMed PMID: 17023859.

8. Sugawara K, Iesato N, Katayose M. Segmental Lordosis of the Spondylolytic Vertebrae in Adolescent Lumbar Spondylolysis: Differences between Bilateral L5 and L4 Spondylolysis. Asian Spine Journal. 2018;12(6):1037-42. Epub 2018/10/17. doi: 10.31616/asj.2018.12.6.1037. PubMed PMID: 30322253; PubMed Central PMCID: PMCPMC6284115.

9. Marty C, Boisaubert B, Descamps H, Montigny JP, Hecquet J, Legaye J, et al. The sagittal anatomy of the sacrum among young adults, infants, and spondylolisthesis patients. European Spine Journal. 2002;11(2):119-25. Epub 2002/04/17. doi: 10.1007/s00586-001-0349-7. PubMed PMID: 11956917; PubMed Central PMCID: PMCPMC3610511.

10. Labelle H, Mac-Thiong JM, Roussouly P. Spino-pelvic sagittal balance of spondylolisthesis: a review and classification. European Spine Journal. 2011;20 Suppl 5(Suppl 5):641-6. Epub 2011/08/03. doi: 10.1007/s00586-011-1932-1. PubMed PMID: 21809015; PubMed Central PMCID: PMCPMC3175928.

11. Sairyo K. Spondylolysis Fracture Angle in Children and Adolescents on CT Indicates the Facture Producing Force Vector : A Biomechanical Rationale. The Internet Journal of Spine Surgery. 2005;1(2):2.

12. Nagamoto H, Abe M, Konashi Y, Kimura R, Takahashi M, Oizumi A. Rotation-related sports players demonstrate rotation-type lumbar spondylolysis fracture angle and decreased hip internal rotation range of motion. Journal of Orthopaedics. 2021;28:101-6. Epub 2021/12/14. doi: 10.1016/j.jor.2021.11.010. PubMed PMID: 34898928; PubMed Central PMCID: PMCPMC8640617.

13. Matsumoto ME, Czerniecki JM, Shakir A, Suri P, Orendurff M, Morgenroth DC. The relationship between lumbar lordosis angle and low back pain in individuals with transfemoral amputation. Prosthetics and Orthotics International. 2019;43(2):227-32. Epub 2018/08/21. doi: 10.1177/0309364618792746. PubMed PMID: 30122108.

14. Terai T, Sairyo K, Goel VK, Ebraheim N, Biyani A, Faizan A, et al. Spondylolysis originates in the ventral aspect of the pars interarticularis: a clinical and biomechanical study. The Journal of Bone and Joint Surgery. 2010;92(8):1123-7. Epub 2010/08/03. doi: 10.1302/0301-620X.92B8.22883. PubMed PMID: 20675758.

 

Reviewer #2:

1　The authors should more analyze the absence of difference regarding SSA between the groups, especially between the unilateral fracture group and the bilateral fracture group.

→We have added the discussion (Line 151- ).

There were no significant differences in the SSA between the spondylolysis and control groups, and between bilateral and unilateral groups. Lumbar lordosis is related to the alignment of the upper spine and pelvis. Therefore, LLA in the bilateral group may be related to the upper spine.

2　Did the measurements of the LLA and SSA have been carried out on standing position? the authors should precise it.

→Thank you for asking this question. X-ray films were taken in side-lying position. This content had been described in the Materials and Methods section of the revised manuscript (Lines 72- ).

Inclusion criteria were as follows: patients who had computed tomography (CT) and X-ray films in the side-lying position for the initial visit, and patients with progressive or terminal stage [12].

3　Figure 3 should be more explained because the graphic remains unclear.

→We have revised this legend appropriately (Lines 141- ).

Fig 3. Scatter plot of spondylolysis fracture angle (SFA) and lumbar lordosis angle (LLA) in bilateral spondylolysis (A) and unilateral spondylolysis (B).

There was no significant correlation between SFA and LLA [bilateral (p = .072), unilateral (p = .76)].

4　In the discussion part, the authors should more argue their hypothesis as concern the occurrence of isthmic unilateral fracture rather than bilateral isthmic fracture. It would be possibly relevant to add a schematic figure explaining the relationship between SFA, uni or bilateral isthmic fracture and type of sport movement leading to isthmic stress fracture.

→This was a retrospective study and the relationship between the fracture site and sports movement was not examined. Therefore, we were not able to discuss this in depth and create a schematic figure.

---

## [Decision Letter · Decision Letter 1]

22 Sep 2022

PONE-D-22-11206R1Comparison of alignment and spondylolysis fracture angle in bilateral and unilateral spondylolysisPLOS ONE

Dear Dr. Matsuzawa,

Thank you for submitting your manuscript to PLOS ONE. After careful consideration, we feel that it has merit but does not fully meet PLOS ONE’s publication criteria as it currently stands. Therefore, we invite you to submit a revised version of the manuscript that addresses the points raised during the review process.

We look forward to receiving your revised manuscript.

Kind regards,

Walid Kamal Abdelbasset, Ph.D.

Academic Editor

PLOS ONE

Journal Requirements:

Reviewers' comments:

Reviewer's Responses to Questions

**Comments to the Author**

1. If the authors have adequately addressed your comments raised in a previous round of review and you feel that this manuscript is now acceptable for publication, you may indicate that here to bypass the “Comments to the Author” section, enter your conflict of interest statement in the “Confidential to Editor” section, and submit your "Accept" recommendation.

Reviewer #1: All comments have been addressed

2. Is the manuscript technically sound, and do the data support the conclusions?

Reviewer #1: Yes

3. Has the statistical analysis been performed appropriately and rigorously? 

Reviewer #1: Yes

4. Have the authors made all data underlying the findings in their manuscript fully available?

Reviewer #1: Yes

5. Is the manuscript presented in an intelligible fashion and written in standard English?

Reviewer #1: Yes

6. Review Comments to the Author

Reviewer #1: Line 147: LLA was significantly larger

Line 149: Increase in LLA

Have already mentioned the above issue in the initial review.

Besides the above mentioning, the authors have answered and revised thoroughly and worth publication in the journal.

7. PLOS authors have the option to publish the peer review history of their article (what does this mean?). If published, this will include your full peer review and any attached files.

Reviewer #1: No

---

## [Author Response · Author response to Decision Letter 1]

23 Sep 2022

September 23, 2022

Editorial Board

PLOS ONE

Manuscript Number: PONE-D-22-11206

“Comparison of alignment and spondylolysis fracture angle in bilateral and unilateral spondylolysis” by Kanta Matsuzawa

Dear Editor:

Thank you for your letter regarding our submission. We are grateful for the detailed feedback provided by the reviewers, which we feel has helped us significantly improve the paper. Attached are our point-by-point responses to the reviewers’ comments and our revised manuscript, which we hope will now meet your approval. We believe that our revisions have addressed the issues raised by the reviewers and trust that the manuscript is now suitable for publication in PLOS ONE. Thank you again for your thoughtful comments, and we look forward to hearing from you soon.

Sincerely,

Kanta Matsuzawa 

Journal Requirements:

→We have not changed the references. We modified only the format of references.

1. Sakai T, Sairyo K, Suzue N, Kosaka H, Yasui N. Incidence and etiology of lumbar spondylolysis: review of the literature. Journal of Orthopaedic Science. 2010;15:281-288. doi: 10.1007/s00776-010-1454-4, PubMed PMID: 20559793.

2. Randall RM, Silverstein M, Goodwin R. Review of pediatric spondylolysis and spondylolisthesis. Sports Medicine and Arthroscopy Review. 2016;24:184-187. doi: 10.1097/JSA.0000000000000127, PubMed PMID: 27811518.

3. Grazina R, Andrade R, Santos FL, Marinhas J, Pereira R, Bastos R, et al. Return to play after conservative and surgical treatment in athletes with spondylolysis: A systematic review. Physical Therapy in Sport : official journal of the Association of Chartered Physiotherapists in Sports Medicine. 2019;37:34-43. doi: 10.1016/j.ptsp.2019.02.005, PubMed PMID: 30826586.

4. Cooper G. Non-operative treatment of the lumbar spine. 2015.

5. Sairyo K, Katoh S, Sasa T, Yasui N, Goel VK, Vadapalli S, et al. Athletes with unilateral spondylolysis are at risk of stress fracture at the contralateral pedicle and pars interarticularis: a clinical and biomechanical study. The American Journal of Sports Medicine. 2005;33(4):583-590. doi: 10.1177/0363546504269035, PubMed PMID: 15722292.

6. El Rassi G, Takemitsu M, Glutting J, Shah SA. Effect of sports modification on clinical outcome in children and adolescent athletes with symptomatic lumbar spondylolysis. American Journal of Physical Medicine and Rehabilitation. 2013;92:1070-1074. doi: 10.1097/PHM.0b013e318296da7e, PubMed PMID: 24141103.

7. Roussouly P, Gollogly S, Berthonnaud E, Labelle H, Weidenbaum M. Sagittal alignment of the spine and pelvis in the presence of L5-s1 isthmic lysis and low-grade spondylolisthesis. Spine (Phila Pa 1976). 2006;31(21):2484-2490. doi: 10.1097/01.brs.0000239155.37261.69, PubMed PMID: 17023859.

8. Sugawara K, Iesato N, Katayose M. Segmental lordosis of the spondylolytic vertebrae in adolescent lumbar spondylolysis: differences between bilateral L5 and L4 spondylolysis. Asian Spine Journal. 2018;12:1037-1042. doi: 10.31616/asj.2018.12.6.1037, PubMed PMID: 30322253; PubMed Central PMCID: PMC6284115.

9. Marty C, Boisaubert B, Descamps H, Montigny JP, Hecquet J, Legaye J, et al. The sagittal anatomy of the sacrum among young adults, infants, and spondylolisthesis patients. European Spine Journal. 2002;11(2):119-125. doi: 10.1007/s00586-001-0349-7, PubMed PMID: 11956917; PubMed Central PMCID: PMC3610511.

10. Labelle H, Mac-Thiong JM, Roussouly P. Spino-pelvic sagittal balance of spondylolisthesis: a review and classification. European Spine Journal. 2011;20 Suppl 5:641-646. doi: 10.1007/s00586-011-1932-1,PubMed PMID: 21809015; PubMed Central PMCID: PMC3175928.

11. Sairyo K. Spondylolysis fracture angle in children and adolescents on CT indicates the facture producing force vector : A biomechanical rationale. The Internet Journal of Spine Surgery. 2005;1:2.

12. Nagamoto H, Abe M, Konashi Y, Kimura R, Takahashi M, Oizumi A. Rotation-related sports players demonstrate rotation-type lumbar spondylolysis fracture angle and decreased hip internal rotation range of motion. Journal of Orthopaedics. 2021;28:101-106. doi: 10.1016/j.jor.2021.11.010, PubMed PMID: 34898928; PubMed Central PMCID: PMC8640617.

13. Matsumoto ME, Czerniecki JM, Shakir A, Suri P, Orendurff M, Morgenroth DC. The relationship between lumbar lordosis angle and low back pain in individuals with transfemoral amputation. Prosthetics and Orthotics International. 2019;43:227-232. doi: 10.1177/0309364618792746, PubMed PMID: 30122108.

14. Terai T, Sairyo K, Goel VK, Ebraheim N, Biyani A, Faizan A, et al. Spondylolysis originates in the ventral aspect of the pars interarticularis: a clinical and biomechanical study. The Journal of Bone and Joint Surgery. 2010;92(8):1123-1127. doi: 10.1302/0301-620X.92B8.22883, PubMed PMID: 20675758.

 

Reviewer #1:

1　Line 147: LLA was significantly larger. Line 149: Increase in LLA. Have already mentioned the above issue in the initial review. Besides the above mentioning, the authors have answered and revised thoroughly and worth publication in the journal.

→Thank you for the suggestion. We have revised as suggested (Lines 147- ).

We measured LLA and SSA as alignment; LLA was significantly larger in all the spondylolysis patients than in the control group, and the LLA was significantly larger in bilateral than unilateral spondylolysis. Increase in LLA may lead to an increase in compressive and shearing forces on the pars interarticularis with lumbar extension [8].

---

## [Decision Letter · Decision Letter 2]

5 Oct 2022

Comparison of alignment and spondylolysis fracture angle in bilateral and unilateral spondylolysis

PONE-D-22-11206R2

Dear Dr. Matsuzawa,

We’re pleased to inform you that your manuscript has been judged scientifically suitable for publication and will be formally accepted for publication once it meets all outstanding technical requirements.

Kind regards,

Walid Kamal Abdelbasset, Ph.D.

Academic Editor

PLOS ONE

Additional Editor Comments (optional):

Reviewers' comments:

Reviewer's Responses to Questions

**Comments to the Author**

1. If the authors have adequately addressed your comments raised in a previous round of review and you feel that this manuscript is now acceptable for publication, you may indicate that here to bypass the “Comments to the Author” section, enter your conflict of interest statement in the “Confidential to Editor” section, and submit your "Accept" recommendation.

Reviewer #1: All comments have been addressed

2. Is the manuscript technically sound, and do the data support the conclusions?

Reviewer #1: Yes

3. Has the statistical analysis been performed appropriately and rigorously? 

Reviewer #1: Yes

4. Have the authors made all data underlying the findings in their manuscript fully available?

Reviewer #1: Yes

5. Is the manuscript presented in an intelligible fashion and written in standard English?

Reviewer #1: Yes

6. Review Comments to the Author

Reviewer #1: I suppose the authors have responded well to the reviewer's questions and is now suitable for publication.

7. PLOS authors have the option to publish the peer review history of their article (what does this mean?). If published, this will include your full peer review and any attached files.

Reviewer #1: No

---

## [Editor Report · Acceptance letter]

7 Oct 2022

PONE-D-22-11206R2 

Comparison of alignment and spondylolysis fracture angle in bilateral and unilateral spondylolysis 

Dear Dr. Matsuzawa:

I'm pleased to inform you that your manuscript has been deemed suitable for publication in PLOS ONE. Congratulations! Your manuscript is now with our production department. 

Kind regards, 

on behalf of

Dr. Walid Kamal Abdelbasset 

Academic Editor

PLOS ONE